# Rapid endogenic rock recycling in magmatic arcs

Jun-Yong Li[1,2], Ming Tang[2,3], Cin-Ty A. Lee [2], Xiao-Lei Wang [1✉], Zhi-Dong Gu[4], Xiao-Ping Xia [5], Di Wang[1], De-Hong Du[1] & Lin-Sen Li[1]

In subduction zones, materials on Earth's surface can be transported to the deep crust or mantle, but the exact mechanisms and the nature of the recycled materials are not fully understood. Here, we report a set of migmatites from western Yangtze Block, China. These migmatites have similar bulk compositions as forearc sediments. Zircon age distributions and Hf–O isotopes indicate that the precursors of the sediments were predominantly derived from juvenile arc crust itself. Using phase equilibria modeling, we show that the sediments experienced high temperature-to-pressure ratio metamorphism and were most likely transported to deep arc crust by intracrustal thrust faults. By dating the magmatic zircon cores and overgrowth rims, we find that the entire rock cycle, from arc magmatism, to weathering at the surface, then to burial and remelting in the deep crust, took place within ~10 Myr. Our findings highlight thrust faults as an efficient recycling channel in compressional arcs and endogenic recycling as an important mechanism driving internal redistribution and differentiation of arc crust.

---

[1] State Key Laboratory for Mineral Deposits Research, School of Earth Sciences and Engineering, Nanjing University, Nanjing, China. [2] Department of Earth, Environmental and Planetary Sciences, Rice University, Houston, TX, USA. [3] School of Earth and Space Sciences, Peking University, Beijing, China. [4] Research Institute of Petroleum Exploration and Development, Beijing, China. [5] State Key Laboratory of Isotope Geochemistry, Guangzhou Institute of Geochemistry, Chinese Academy of Sciences, Guangzhou, China. ✉email: wxl@nju.edu.cn

Magmatic arcs witness the interplay between endogenous and exogenous processes, including magmatism, crustal thickening, uplift, erosion, sedimentation and burial of detritus[1–4]. Magmatism produces new crust, which later interacts with the hydrosphere and atmosphere through erosion and weathering. On the other hand, crustal materials from the surface are recycled to Earth's interior. This chain of processes in magmatic arcs play important roles in driving much of the mass exchange between Earth's interior and surface. The inward transport of surface materials, including volatiles, has profound influence on the cycling of carbon, oxygen, sulfur, etc. on Earth's surface and may alter the chemical and physical properties of the deep crust and even mantle.

Nearly every Phanerozoic arc in the world exhibit crustal signatures in geochemistry, suggesting pervasive crustal recycling in the formation of arc crust. Conventional views link crustal recycling processes to slab subduction, including sediment subduction and subduction erosion (± relamination) have been widely invoked to explain the crustal signatures seen in most arc magmas[5,6]. Yet the recent work on continental arcs hints at thrust faults as potential recycling channels[7–10].

Here, we examined a suite of migmatites from a Neoproterozoic magmatic arc in western China. We used combined petrologic, geochronologic and geochemical studies of these samples to understand the nature of the recycled materials and evaluate how thrust faults may contribute to rock recycling in compressional arc settings.

## Results

**Geological setting and samples.** The Yangtze Block in Eastern Asia consists of Archean–Paleoproterozoic crystalline basement surrounded by Neoproterozoic fold belts. It is bounded by the Tibetan Plateau to the west, the North China Block to the north and the Cathaysia Block to the southeast. It was placed in a marginal position in Rodinia supercontinent and has underwent a long-term evolution and complex tectonic-magmatic processes in a continental margin setting during Neoproterozoic[11–13]. The western margin of the Yangtze Block became tectonically active since the early Neoproterozoic; it started with intra-oceanic arc magmatism before $971 \pm 16$ Ma (ref. [14]) and then transitioned to Andean-type magmatism at ca. 870 Ma (ref. [15]). This ancient subduction relic was lately imaged by deep seismic reflection profile[16]. The prolonged magmatic history gave rise to linearly distributed Neoproterozoic arc magmatic rocks spanning over 800 km (Supplementary Fig. 1a). The Longmenshan Thrust Belt to the northwestern margin of the Yangtze Block exposed abundant Neoproterozoic plutonic complexes due to the major Miocene extrusion and thrust process[17], of which the largest one is known as the Pengguan Complex, comprising voluminous 860–750 Ma plutonic rocks (Supplementary Fig. 1b, c). The Huangshuihe Group in the core region of the Pengguan Complex exists as a huge roof pendant of the plutonic rocks and consists of metamorphic rocks of schist, metapelite, quartzite and meta-pyroclastic rock. Ductile deformation, faults, mylonite with S-C fabric, and migmatitic lineation are extensive in the sequences.

Two main types of migmatites were identified in the field (Supplementary Fig. 2): the inhomogeneous migmatites (or diatexite) contain abundant blocks of melanosome and associated aplite vein; the stromatic migmatites preserve a regular layered structure and are characterized by centimeter-thick, foliation-parallel leucosome, melanosome and mesosome. Patch-shaped neosomes are abundant in stromatic migmatites and formed during incipient partial melting. Large leucosomes (~50 cm in width) occur occasionally and are usually fed by a few small leucosome veins. The stromatic migmatites have a NNW-dipping foliation ($S_1$) (~355°/48°) defined by oriented biotite or feldspar augen. The $S_1$ foliation is parallel to bedding planes defined by the metapelite (Supplementary Fig. 2c) and is folded locally by syn-anatectic deformation on varying scale (Supplementary Fig. 2e, f). The fold axial planes ($S_2$) generally display E–W striking, S-dipping orientation. Besides, the study area was superimposed by massive high-angle, S-verging thrust faults (Supplementary Fig. 2g), which should be linked with post-Mesozoic structural tectonics[17].

Six stromatic migmatite and one leucosome samples in the Huangshuihe Group were collected in this study (Supplementary Fig. 1d). The main minerals in migmatite are plagioclase, biotite, K-feldspar, quartz and muscovite (Supplementary Fig. 3). Anatexis of primary mineral assemblage led to prevalent zircon overgrowth and muscovite-rimmed biotite in the migmatite (Supplementary Figs. 3 and 5–7). Entrainment of peritectic phase, which consists of small spessartine-rich garnet grains, biotite, muscovite, quartz, plagioclase, K-feldspar and Fe-oxides, was found in 16YX-1-1 (Supplementary Fig. 3 and Supplementary Data 4). The reaction of "biotite + MnO, $Al_2O_3$, $SiO_2$ (from melt) = garnet + muscovite"[18] may control garnet paragenesis. These observations are indicative of near-solidus partial melting with local melt segregation.

**Zircon U–Pb–Hf–O isotopes.** Most zircon grains in the Pengguan migmatites have core-rim structures. The zircon core domains, presumably derived from arc magmatic detritus, show limited variation in their ages, concentrating at ~830–870 Ma, with few at ~930 Ma (Fig. 1a), and have mantle-like to slightly elevated $\delta^{18}O$ values (5.3 to 7.4‰) (Fig. 1b). Their $\varepsilon_{Hf}(t)$ values vary from −3 to +13, with most being positive, indicative of heterogeneous but generally juvenile sources. Zircon overgrowth rims are slightly younger than the maximum depositional age for each sample, with U − Pb dates generally ranging from ~815 Ma to ~860 Ma (Supplementary Data 3). The overgrowth rims have significantly higher $\delta^{18}O$ values (9.3 to 13.3‰) compared with those of core domains, despite their similar $\varepsilon_{Hf}(t)$ range (−3 to +8 except one analysis of −9) as core domains (Supplementary Data 2). Zircon grains from the leucosome sample show homogeneous $\delta^{18}O$ values (11.1 to 13.4‰) with a large range of $\varepsilon_{Hf}(t)$ values (−6.9 to +8.4) (Supplementary Data 2). All $\varepsilon_{Hf}(t)$ values were calculated to $t = 850$ Ma in order to facilitate comparison.

**Anatexis P-T conditions.** We reconstructed the metamorphic P-T conditions for the Pengguan migmatites using Perple_X 6.9.0 (http://www.perplex.ethz.ch). The bulk rock composition of sample 16YX-1-1 was chosen for calculation because this sample clearly documents: (1) mineral-melt interaction; (2) coexistence of minerals (Mn-rich garnet + biotite + muscovite + quartz + plagioclase + K-feldspar + Fe-oxides) and; (3) minor partial melting with no evident melt migration. In the calculated P-T pseudosection, the mineral assemblage of the Pengguan migmatite falls in a narrow domain (domain 1 in Fig. 2a) near the solidus. Using Si pfu in muscovite from 16YX-1-1 (3.08 to 3.14, in moles per formula unit; Supplementary Data 4), which is sensitive to pressure in the K-feldspar + phlogopite + quartz system[19], we further constrained the anatexis P–T conditions to ~670 °C and 5.9 − 8.1 kbar (Fig. 2). The low anatexis temperature is also consistent with the extremely low Th/U ratios of the zircon overgrowth rims (Fig. 1a). At near-solidus temperatures, Th concentration in the melt is largely buffered by Th-rich accessory minerals (such as monazite and allanite)[20].

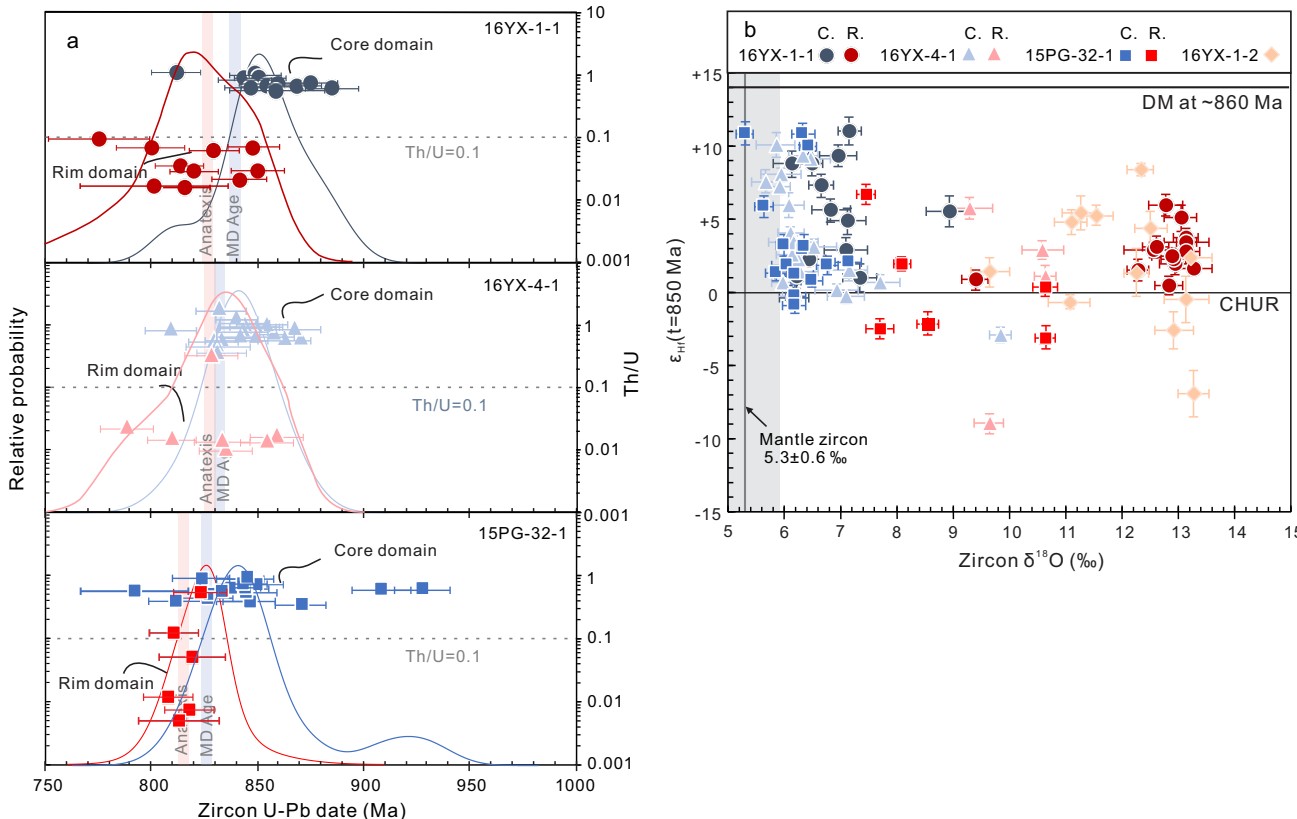

**Fig. 1 Zircon U−Pb age distributions and zircon δ¹⁸O-ε_Hf(t) values of the Pengguan migmatites. a** Core and rim age distribution of zircon grains extracted from Pengguan migmatites. Unimodal core and rim zircon U-Pb dates distribution is shown in the diagram. Anatexis and maximum depositional ages (MD Age, see discussion for the calculation) of each sample are marked with light red and blue, respectively. Zircon U−Pb dates with concordant U−Th−Pb isotopes were used here. Error bars are shown at one standard error. The concordant criteria were adopted from Spencer et al.[40]. **b** δ¹⁸O-ε_Hf(t = 850 Ma) diagram for zircon core and rim domains of three migmatite samples and zircon grains from leucosome. Error bars are shown at one standard error for δ¹⁸O value and two standard errors for ε_Hf(t) value. C = core domain, R = rim domain. Mantle zircon δ¹⁸O = 5.3 ± 0.6‰ (ref.[41]).

## Discussion

The Pengguan migmatites are peraluminous with aluminum saturation indices (ASI) of 1.10–1.44 (Supplementary Data 1). Muscovite and peritectic garnets are observed in all samples, indicating peraluminous composition of the protoliths. These migmatites also show geochemical signatures similar to those arc magmatic rocks and forearc sediments from Peninsular Ranges batholith, but distinct from those of MORB and intraplate volcanics (Fig. 3), suggesting that the protoliths of these migmatites are dominated by arc-related magmatic detritus. This view is also consistent with the observation that the magmatic cores of zircon in the migmatites have very similar age distributions to that of the arc-related magmatic rocks in the study area (Fig. 1a). The absence of pre-Neoproterozoic zircon xenocrysts hints that forearc magmatic detritus may have served as the protoliths of the Pengguan migmatite. The consistent and juvenile Hf isotopes of the zircon cores and overgrowth rims lends further support for the arc origin of the migmatite precursor materials. We thus suggest the Pengguan migmatites documented a process that recycled the arc crust itself, and we refer to this process as endogenic recycling to distinguish it from recycling of oceanic sediments or oceanic crust into subduction zones.

We note that the zircon overgrowth rims have systematically higher δ¹⁸O values than the magmatic cores (Fig. 1b), which is indicative of equilibrium with high-δ¹⁸O anatectic melts during crystallization. High δ¹⁸O is a diagnostic signature of low-temperature water-rock interaction at Earth's surface. Thereby the protoliths of the Pengguan migmatites must have undergone

some extent of chemical weathering and O isotope exchange at low temperatures before being buried and remelted. Downward infiltration of meteoric water may be another important mechanism to introduce oxygen isotopic heterogeneity to the deep crust. But this mechanism would likely cause water-rock interaction at high temperatures and impart low δ¹⁸O signature to the rocks, as has been clearly seen in the lower oceanic crust[21]. In addition, the maximum penetration depth of meteoric water ranges from 5 to 18 km (ref.[22]), which is less than the depth of anatexis (~18 to 24 km) calculated for our migmatites. We thus exclude interaction with downward infiltrated meteoric water as a likely mechanism to explain the high δ¹⁸O recorded by the zircon rims of this study.

An important question pertains to how the magmatic detritus that had been initially deposited at the surface was transported to the hot deep crust. In magmatic arc settings, recycling of surface rocks has generally been associated with slab subduction. Subducting slabs can directly bring trench sediments to the deep crust or even mantle[5,6]. Subduction erosion has also been recognized as an important mechanism for downward transport of shallow crustal materials[5]. Slab tops are cold (dT/dP = <34 °C/kbar) (Fig. 2b; estimated from Peacock[23]) and melting of the sediments deposited at the slab surface is generally considered difficult at crustal depths[24]. Phase equilibrium modeling shows that the Pengguan migmatites formed at ~670 °C and 5.9−8.1 kbar. These P–T conditions translate into a hot geothermal gradient of 83−114 °C /kbar or 25 to 34 °C /km, considerably hotter than slab top geothermal gradients but consistent with those seen

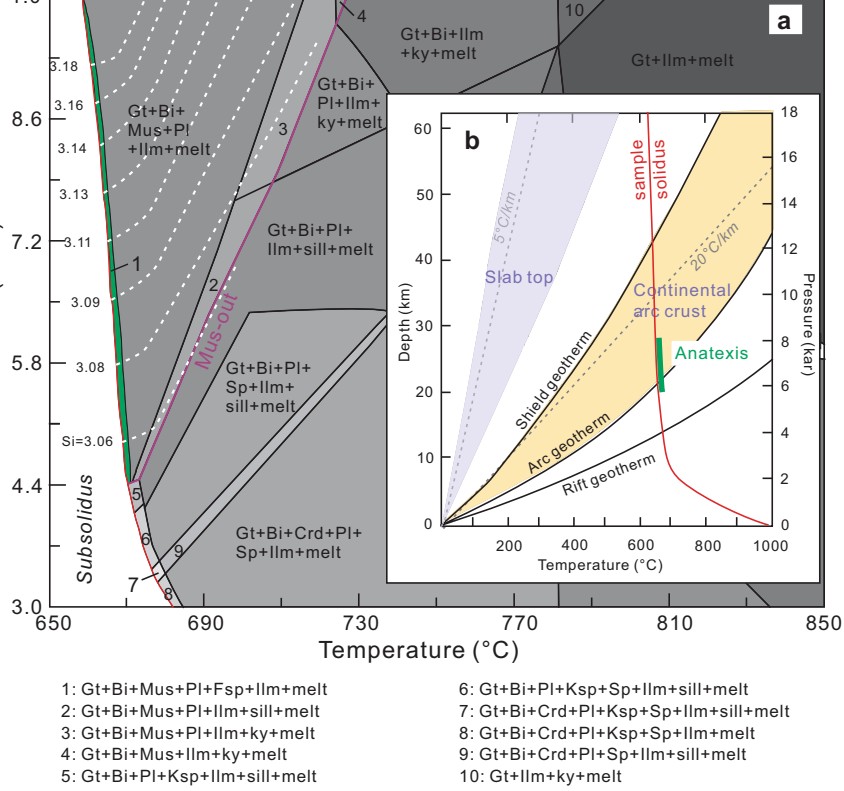

**Fig. 2 Reconstructed anatexis P–T conditions for the Pengguan migmatites. a** P–T pseudosection calculated for the Pengguan migmatite (16YX-1-1) in the MnNKCFMASH system (Quartz and $H_2O$ in excess). Internally consistent thermodynamic dataset of Holland and Powell[42] was used. Mineral assemblage in the thin green belt along the solidus is consistent with the mineral composition and field observations of sample 16YX-1-1. The predicted Si isopleths content in muscovite (3.06–3.18, in molar per formula unit) are shown by the white dotted lines. Gt–garnet, Bi–biotite, Mus–muscovite, Pl–plagioclase, Fsp–K-feldspar, Ilm–ilmenite, sil–sillimanite, ky–kyanite, Sp–spinel, Crd–cordierite. **b** P–T conditions of the Pengguan migmatite formation (green line) projected on geothermal gradients in various geologic settings (modified after Rothstein and Manning[25], Hopkins et al.[43] and Peacock[23]).

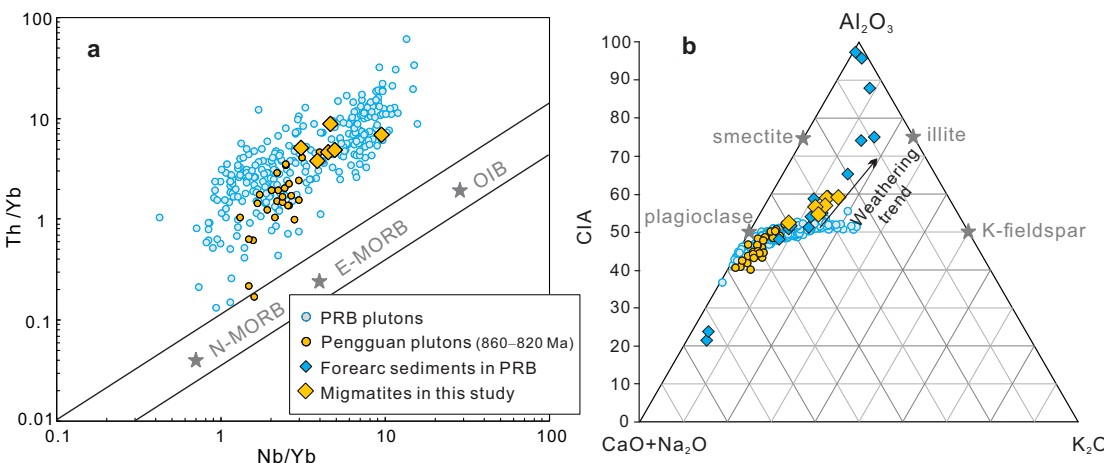

**Fig. 3 Geochemical affinity of the Pengguan migmatites compared with those of 860–820 Ma plutonic rocks nearby, the plutonic rocks from Peninsular Ranges batholith (PRB)[44] and forearc sediments derived from PRB[45]. a** The Nb/Yb-Th/Yb discrimination diagram is after Pearce[46]. **b** The A–CN–K–CIA (Chemical index of alteration) diagram is after Jiang and Lee[45]. OIB – Ocean-island basalt; E- and N-MORB – enriched and normal mid-ocean-ridge basalt. Data sources are provided in Supplementary Data 1.

in arc crust with continuous magmatic inflation[25] (Fig. 2b). This would imply that the Pengguan forearc detrital sediments, shortly after their deposition, were rapidly transported to the deep crust beneath the active arc volcanic front. We suggest the most likely recycling mechanism is via deep thrust faults in the upper continental plate rather than by slab subduction (Fig. 4). Downward

flow of wall rocks during magma ascent[26] could be another mechanism in transporting surface materials to the deep crust, but we think it less likely occurred because: (1) wall rock xenoliths were not seen in the plutonic rocks, and (2) vertical flow foliation or lineation is absent in the wall rocks. In compressional magmatic arcs, including mature island arcs and continental arcs, fold

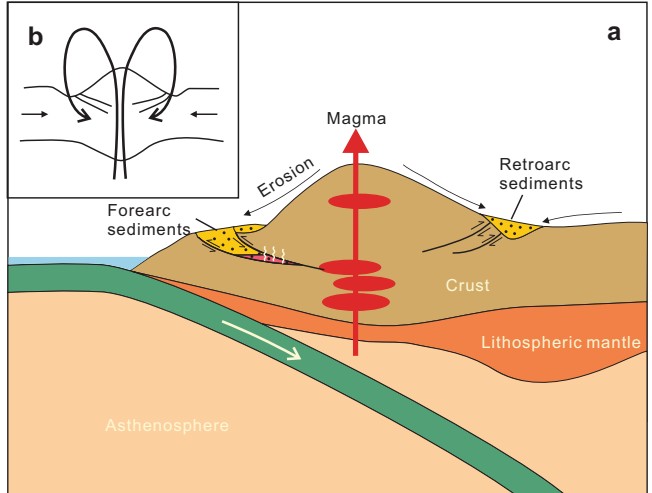

**Fig. 4 Cartoon showing rapid endogenic recycling of arc magmatic rocks through thrust channels in continental arcs (not to scale). a** Thrust-driven rock recycling in this study: from arc magmatism, to erosion and weathering at the surface, to forearc sedimentation, then to burial and remelting. **b** Sketch diagram illustrating the self-recycling process in arc system.

and thrust belts may extensively develop in the forearc and inboard side of the arc and serve as important crustal recycling channels. Typical examples include the thrust fault systems in Lachlan orogen[27], Japan arc[28] and the Cordilleran continental arc system[7].

The nearly identical age distributions of the zircon cores and overgrowth rims (Fig. 1a) hint at a fast rock cycle, from arc magmatism to water-rock interaction, then to burial and remelting. To estimate the timescale and rate of burial for the Pengguan forearc magmatic detritus, we took the weighted average value of the 50% of youngest $U-Pb$ dates with concordant $U-Th-Pb$ isotopes from zircon core domains as the maximum depositional age, and the weighted average age of zircon rims as the remelting (anatexis) age (Supplementary Data 2 and Fig. 1a). In doing so, we see that the maximum depositional ages are less than 1 to 14 Myr older than the remelting ages for each sample. With the errors of zircon dating taken into account, the magmatic protoliths of the Pengguan forearc sediments must have been exhumed, deposited in a sedimentary basin, and then buried to the depth of crustal anatexis on a ~10 Myr timescale. This would imply an efficient burial process with minimum burial rate of 2–3 mm/y.

Our findings point to endogenic recycling as an important mechanism driving internal redistribution of arc crustal materials. The role of such endogenic recycling in the formation of arc crust has been largely overlooked in the past. Radiogenic isotopes are widely employed to constrain crustal recycling processes, but given the short residence time (e.g., 10 Myr), radiogenic isotopes can be completely blind to endogenic recycling. We speculate that extensive endogenic recycling may also generate significant decoupling between radiogenic and stable isotope compositions in recycled materials and their derivative melts, which may further complicate the use of isotope-based proxies in tracing crustal recycling in arc settings.

Rapid endogenic recycling may be facilitated by thrust fault networks. Deep thrust faults may serve as critical transport channels connecting the surface and deep arc crust with ongoing magmatism. And by transporting hydrated surface crustal materials to the deep crust, endogenic recycling enhances the

overall differentiation of arc crust. Because large-scale thrust faults necessarily form in compressional settings, efficient endogenic recycling may partly explain why thick arc crust formed in compressional settings (e.g., continental arcs) tend to be more differentiated than thin arc crust formed in extensional settings (e.g., immature island arcs)[29].

## Methods

**In situ Zircon isotopes.** Zircon grains were separated using conventional density and magnetic techniques, mounted in epoxy resin disk, and polished to expose their internal texture. In situ U–Th–Pb–Hf–O isotope analyses were carried out guided by cathodoluminescence (CL) images and transmitted and reflected photographs. The CL images were taken with a Carl Zeiss Supra 55 field-emission scanning electron microscope (FE-SEM) coupled to a GATAN MonoCL4 detector at the State Key Laboratory for Mineral Deposits Research in Nanjing University (MiDeR-NJU) under following conditions: an accelerating voltage of 3 kV, working distance of 11.5 mm, and aperture size of 30 μm.

Zircon U–Pb isotopes were analyzed using the Cameca IMS-1280HR second ion mass spectrometry (SIMS) at the Guangzhou Institute of Geochemistry, Chinese Academy of Sciences (SKLabBIG GIG CAS) under the following operating conditions: 7 scan cycle, ~8 nA primary $O_2^-$ beam, $20 \times 30$ μm spot size, and ~5400 mass resolving power. Calibration of Pb/U ratios is relative to the primary standard zircon Plešovice[30] and is based on an observed linear relationship between ln ($^{206}$Pb/$^{238}$U) and ln ($^{238}$U$^{16}$O$_2$/$^{238}$U)[31]. A long-term uncertainty of 1.5% (1 RSD) for $^{206}$Pb/$^{238}$U measurements of the standard zircon was propagated to the unknowns, despite that the measured $^{206}$Pb/$^{238}$U error in a specific session is generally around 1% (1 RSD) or less. U and Th concentrations of unknowns were also calibrated relative to the standard zircon Plešovice, with Th and U concentrations of 78 and 755 ppm, respectively[30]. Measured compositions were corrected for common Pb using non-radiogenic $^{204}$Pb. A secondary standard zircon Qinghu[32] was analyzed as unknown to monitor the reliability of the whole procedure. Data reduction was carried out using the Isoplot/Ex 3 software[33]. Eleven measurements of the Qinghu zircon standard during the course of the study yielded a weighted mean $^{238}$U/$^{206}$Pb age of 159 ± 2 Ma (MSWD = 0.72), consistent with its recommended value of 159 ± 0.2 Ma (ref. [32]).

After U–Pb dating, the sample mount was re-ground for ~5 μm to ensure any oxygen implanted in zircon surface from the O$_2^-$ beam used for U–Pb analysis is completely removed. Zircon oxygen isotope analyses were also conducted using SIMS at SKLabBIG GIG CAS. The $^{133}$Cs$^+$ primary ion beam was accelerated at 10 kV, with an intensity of ~2 nA and focused to an area of ϕ 10 μm on the sample surface and the size of analytical spots is about 20 μm in diameter (10 μm beam diameter +10 μm raster). Oxygen isotopes were measured in multi-collector mode using two off-axis Faraday cups. The measured oxygen isotopic data were corrected for instrumental mass fractionation (IMF) using the Penglai zircon standard[34] ($\delta^{18}O_{VSMOW}$ = 5.31 ± 0.1‰), which was analyzed once every four unknowns, using sample-standard bracketing method. The internal precision of a single analysis generally was better than 0.1‰ (1σ) for the $^{18}$O/$^{16}$O ratio. As discussed by Kita et al.[35] and Valley and Kita[36], internal precision for a single spot (commonly < 0.1‰, 1σ) is not a good index of analytical quality for stable isotope ratios measured by SIMS. Therefore, the external precision, measured by the spot-to-spot reproducibility of repeated analyses of the Penglai standard, 0.30‰ (2σ, $n$ = 24) is adopted for data evaluation. The Qinghu zircon was used as secondary zircon standard, and seventeen measurements of the standard yielded a weighted mean value of $\delta^{18}$O = 5.39 ± 0.08‰ (2σ; MSWD = 1.3), consistent with the reported value of 5.4 ± 0.2‰ (2σ)[32].

Zircon Lu–Hf isotopic analyses were conducted using a GeoLas 193 nm laser-ablation system attached to a Neptune (Plus) MC-ICP-MS at MiDeR-NJU. Beam diameter of ~44 μm was preferentially adopted to zircon domain with large size, while ~32 μm beam diameters was adopted to zircon domain with its size < 44 μm. Each diameter-change operation will be followed by analysis of zircon standard to ensure the stability of the experiment. Ablation pulse rate and energy density are 10 Hz and 10.5 J/cm² respectively. The ablation times were 60 s. Helium carrier gas transported the ablated sample from the laser-ablation cell via a mixing chamber to the ICPMS torch. Masses $^{172}$Yb, $^{173}$Yb, $^{175}$Lu, $^{176}$Hf + Yb+Lu, $^{177}$Hf, $^{178}$Hf, $^{179}$Hf, and $^{180}$Hf were measured in Faraday cups; all analyses were carried out in static-collection mode. Hf reference solution JMC475 was analyzed during analytical session to allow normalization of the fundamental mass spectrometer performance. Interference of $^{176}$Yb on $^{176}$Hf has been corrected by measuring the $^{172}$Yb isotope and using $^{176}$Yb/$^{172}$Yb to calculate $^{176}$Yb/$^{177}$Hf. The appropriate value of $^{176}$Yb/$^{172}$Yb was determined by spiking the JMC475 Hf standard with Yb and a $^{176}$Yb/$^{172}$Yb = 0.588596 was used for this correction. Interference of $^{176}$Lu on $^{176}$Hf is corrected by measuring the $^{175}$Lu isotope and using $^{176}$Lu/$^{175}$Lu = 0.02658 to calculate $^{176}$Lu/$^{177}$Hf. The interference corrected $^{176}$Hf/$^{177}$Hf was normalized assuming $^{179}$Hf/$^{177}$Hf = 0.7325 for mass bias correction. Reference zircon Mudtank and 91500 were used to monitor accuracy and precision of Hf isotope ratios and instrumental drift with respect to the Lu/Hf ratios. The obtained $^{176}$Hf/$^{177}$Hf ratios were 0.282295 ± 0.000009 ($n$ = 15; MSWD = 2.9) for 91500, and 0.282487 ± 0.00008 ($n$ = 14; MSWD = 3.6) for Mudtank, and were consistent with

the recommended values[37,38]. The obtained $^{178}$Hf/$^{177}$Hf and $^{180}$Hf/$^{177}$Hf ratios were 1.467207 ± 0.000017 ($n = 13$; MSWD = 1.6) and1.886870 ± 0.000049 ($n = 14$; MSWD = 2.8) for zircon 91500, 1.467216 ± 0.000023 ($n = 13$; MSWD = 4) and 1.886871 ± 0.000038 ($n = 13$; MSWD = 2.1) for zircon Mudtank. The stable $^{178}$Hf/$^{177}$Hf and $^{180}$Hf/$^{177}$Hf ratios overlap at 2σ with recommended values reported by Thirlwall and Anczkiewicz[39].

**Whole-rock geochemistry.** Major elements were analyzed using a Thermo ARL9900XP X–ray fluorescence spectrometer (XRF) at the MiDeR-NJU. The analytical precision is generally better than 2% for all elements. Whole-rock rare earth and other trace elements were analyzed using an ICP-MS (Finnigan MAT–Element II) instrument at MiDeR-NJU. Each sample was precisely weighted 30 mg and then was put into a 15 ml Savillex digestion vessel. After being dissolved by HNO3 and the injection of 1 ml 500 ng/ml internal standard Rh solutions, the samples are ready for analyzing. Analytical precision for most elements by ICP- MS is better than 5%. Major and trace element composition data of the migmatite and leucosome samples are provided in Supplementary Data 1.

**Mineral composition.** The mineral major element compositions were determined using a JEOL 53 JXA-8100 electron probe microanalysis (EPMA) at the MiDeR-NJU. The instrument was operated in wavelength-dispersion mode with a beam diameter of 1–2 μm, a 15 kV accelerating voltage, and a 20 nA beam current. Element peaks and backgrounds were measured for all elements with counting times of 10 and 5. Natural and synthetic standards were used. Detection limits were better than 0.02 wt % for the oxides of most elements. All EPMA data were automatically reduced using the ZAF correction program. Mineral major content results are provided in Supplementary Data 4.

## Data availability
Major and trace element composition data of the migmatite and leucosome samples are provided in Supplementary Data 1. Summary and details of Age-δ$^{18}$O-ε$_{Hf}$(t) results from core and rim zircon of the migmatites are provided in Supplementary Data 2 and 3, respectively. Mineral major content results are provided in Supplementary Data 4. Analytical method and results for zircon trace element are provided in Supplementary Data 5.

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

## Acknowledgements

This work was financially supported by the National Natural Science Foundation of China (42025202), the Fundamental Research Funds for the Central Universities (020614380089) and the Dengfeng Project of Nanjing University to X.L.W., and the scholarship from China Scholarship Council (File No. 201806190157) to J.Y.L.

## Author contributions

J.Y.L. initiated the idea, collected samples, conducted whole-rock and mineral analyses and carried out melting modeling. J.Y.L., M.T., C.-T.A.L and X.L.W. wrote the manuscript. X.L.W. designed the project. X.P.X. contributes to SIMS analyses. Z.D.G., D.W., and D.H.D assisted in sample collection. L.S.L. helped in melting modeling. All authors contributed to data interpretation.

## Competing interests

The authors declare no competing interests.
