## [Peer Review File · Nature Communications]

REVIEWER COMMENTS

Reviewer #1 (Remarks to the Author):

This is an interesting manuscript on the topic of arc cannibalization. The idea is not new but appears to be applied here to an ancient arc, for which very little geologic context is known or provided here. There is a bit of a problem postulating thrust faults without any reference to any actual thrust fault. Strange, right?? I am not even sure we know that much about the nature of the "arc" that operated in the Proterozoic in that part of the world- the geologic background provided here is rather minimalistic.

Basically, we are shown some zircons whose low delta 18O cores are overprinted by high delta 18O rims formed not long after- 10-20 My later. That is the paper. Is there any other possibility to increase the oxygen via meteoric water fluxing at depth? It's marginally possible: some work done in the 1990s on core complex 18O (Greg Holk, others) show that meteoric water can be sent down to great depths along normal faults where it can damage the oxygen isotopic signature of minerals like quartz (and presumably zircon as well).

Lots of similar work on the cannibalization topic has been done in areas that have a better known geologic control, such as the western N American Cordillera (the POR work started in the 1970s -Platt, England, many more; the newer work based on zircons – Jacobson, Grove, Barth, Kidder, Ducea in California, Matzel, Gordon, etc etc in the Pacific NW). Some of it is acknowledged here, some other relevant papers, not. There, hypotheses such as forearc thrusting can be put to test. Incidentally, forearc thrusting is rarely the explanation in north America, and structural evidence for that mechanism is most compelling in the central Andes. That vast literature shows that there are several sources for recycled materials in an arc, forearc "thrusting" being just one of them. Ultrashallow subduction does it too at high rates (centimeters per year, see the Southern Californian example – Pelona etc), thrusting from the backarc as well (at mm/year) and vertical transport within the arc (the downward flow model of Saleeby or Paterson, others) is also an option, not mentioned here. Rate constraints provided here are minimalistic (at least a few mm/yr) so that alone cannot distinguish between different hypotheses. A careful look at inherited, older than the arc zircons (if they exist) could distinguish between fore and back arc origins. Downward flow is hard to estimate without a good geologic context.

Some data presented here don't advance the paper's primary argument. For example, Hf isotopes are for the most part useless here. Perplex modeling is only marginally useful because it has been clearly documented these rocks were last seen as migmatites at high temperatures, so yeah, they were part of a hot geotherm. Some references, like the one to Penniston Dorland on what is a hot or cold (subduction) gradient are weird. That is textbook stuff, if anything you can refer to early classic papers by Peacock or Phil England.

I don't have the time to go through the details of the writing, will only point out that duration in geology is My or Myr as opposed to absolute age which one writes as Ma. I am fine with the writing, anybody can understand the message.

Is this worthy of being published in Nature Comm? Only an editor can judge the appeal of this manuscript to their journal. I liked the paper and would publish it if it were up to me, but would have also liked to see more details from within zircons (where the actual data comes from), such as how big the rims are, do they show metamorphic U/Th, what are Ti temperatures across grains, trace chemistry etc. All the paper's punch rests on investigations within grain (age and oxygen) anyway. There is no geology here and for the record, thrust faults are mapped/inferred at a whole different scale with totally different tools. Thrust faults cannot be deduced from "within grain" studies but certain geochemical constraints on the rock cycle can be unraveled at that scale.

Reviewer #3 (Remarks to the Author):

Dear Editor,

Below is my review of Rapid endogenic rock recycling in magmatic arcs by Wang et al. The paper is very nicely written and provides important new data supporting rapid incorporation of supracrustal materials into magmatic systems. I think the greatest weakness of the paper is the reliance of thrust fault networks on transporting sedimentary material into the "magmatic window". It would be useful to provide some geologic evidence for this phenomena happening in active continental arcs.

Below are a few additional requested revisions that would improve the paper.

1) Zircon methodologies are insufficient and need to be expanded. The ICP-MS data are mostly reported in a palatable fashion however the analytical methodology is not adequately explained in the supplementary material.

a. Although the laboratories have been generating high quality data for many years for the sake of thoroughness and reproducibility, the authors need to provide the 'metadata' associated with their analytical sessions (for example laser energy, gas flows, etc.). There are many specific analytical conditions of ICP-MS instruments that change within each analytical session that are important to report. Specific to the ICP-MS data the authors should follow the recommendations outlined by Horstwood et al. (2016 GGR) with respect to reporting of analytical metadata.

b. The authors should also provide information regarding the stable isotope ratios for the zircon Hf data ($^{180}\text{Hf}/^{177}\text{Hf}$ and $^{179}\text{Hf}/^{177}\text{Hf}$) to demonstrate the instrument was working properly. Without these it is impossible to determine whether the analyses are valid. This is common practice and these data should be provided. To make the authors lives easier, they can simply follow the workflow provided by Spencer et al. (GSF 2019).

c. The authors should report the value, uncertainty, and MSWD of the secondary standard material for UPb and Hf analyzed to confirm correct normalization. The authors say they used three different standards but did not indicate which standard is the primary (self-normalized) and secondary (used to assure normalization is accurate).

2) It is not clear how the authors define concordance. Many workers use an arbitrary discordance filter of 10% discordance, but this is unscientific and if the authors used an arbitrary discordance filter they need to justify this with a sensitivity analysis or alternatively they can simply use the uncertainty of the analysis to define whether an analysis is statistically "concordant" or "discordant". Spencer et al. (2016 Geoscience Frontiers) evaluate the issue of discordance in conjunction with uncertainty. This is simply done by evaluating whether a given analysis is within uncertainty of the concordia. If an analysis has a centroid that is 'discordant' but overlaps the concordia within uncertainty, then there is a probability the analysis is actually concordant despite centroid discordance. I urge the authors to re-evaluate the issue of discordance and would recommend against publication of an arbitrary discordance distinction.

3) As a minor point, zircon is a mineral species names and should not be used in the plural sense. The singular is used when writing about a mineral (e.g. titanite) is that it is formally considered a species. Species names are not pluralized in English. For example, *Vulpes vulpes* = red fox. And while we might refer to "foxes in the forest", we wouldn't refer to "*Vulpes vulpes-es* in the forest". Confusion can arise when a common mineral name is also a mineral group name. In contrast "garnets" can refer to different garnet species [spessartine, pyrope, etc.].

Reviewer #1 (Remarks to the Author):

This is an interesting manuscript on the topic of arc cannibalization. The idea is not new but appears to be applied here to an ancient arc, for which very little geologic context is known or

provided here. There is a bit of a problem postulating thrust faults without any reference to any actual thrust fault. Strange, right?? I am not even sure we know that much about the nature of the “arc” that operated in the Proterozoic in that part of the world- the geologic background provided here is rather minimalistic.

Thank you very much for the constructive comments, based on which the geologic context in the revised manuscript has been expanded substantially. There are some debates on the Paleo- to Meso- Proterozoic “arc” activities on Earth, but Neoproterozoic modern plate tectonics and related arc magmatism have been extensively studied and have a lot of geological evidence. The Yangtze Block is just one of the distinct areas behaving Neoproterozoic arc magmatism and crustal accretion as supported by a series of petrological, geochemical and geophysical work. These geological observations led some authors suggest the Yangtze Block placed a marginal position of Rodinia and underwent long-timescale, continent-margin, tectonic-magmatic processes during Neoproterozoic (e.g., Zhao et al., 2011; Cawood et al., 2018). Shu et al. (2021) presented a review on the Neoproterozoic tectono-magmatic processes. The main evidence for the Neoproterozoic subduction and arc accretion includes:

- (1) the occurrence of 866 ± 14 Ma blueschist in the central part of the Jiangnan orogen (Shu et al., 1994);
- (2) the occurrences of ca. 1.0-0.83 Ga ophiolite suites in the eastern part of the Jiangnan orogen (Sun et al., 2018 and references therein) and ca. 1.0 Ga ophiolite suite on the western margin of the Yangtze Block (Li et al., 2021 and references therein);
- (3) the linear distribution of Neoproterozoic igneous rocks along the western and southeastern margins of the Yangtze Block;
- (4) the calc-alkaline and “arc-like” geochemical and isotopic signatures of the Neoproterozoic igneous rocks (e.g., Wang et al., 2004, 2012; Zhao et al., 2018);
- (5) the Neoproterozoic subducted slab relic was lately imaged by deep seismic reflection profile (Gao et al., 2016);
- (6) the Neoproterozoic (880-750 Ma) igneous rocks recorded the orogenic cycle from arc-continent collision, post-collisional extension to post-orogenic extension (Wang et al., 2008, 2012, 2014; Zheng et al., 2008; Zhao et al., 2018).

The study area is located at the Longmenshan Thrust Belt (NW Yangtze Block margin), which displays clear superimposed deformation structures and develops extensive southeast-verging, basement-involved folds and thrust faults. The Neoproterozoic arc structure was largely overprinted by multi-episode younger orogenic processes including a Late Triassic compressional event and a Cenozoic deformation event related to the India-Asia collision (Jia et al., 2006; Wang and Meng, 2009). Therefore, it is difficult and we should be very careful to find actual tectonic evidence for Neoproterozoic thrust systems.

When revising this paper, we took your comments seriously and carried out another field trip to carefully explore structural evidence for the Neoproterozoic thrust. Fortunately, the syn-anatexis deformation can be distinguished from Phanerozoic deformation and records the Neoproterozoic kinematics process. As shown by the Supplementary Fig. 2, the studied migmatites are mainly metatexites exhibiting a regular layered structure and are intruded by Neoproterozoic arc diorites. We found that these migmatites have NNW-dipping S_1 foliation (mean: $355^\circ/48^\circ$) paralleled to the

bedding planes (S_0). The S_1 foliation is locally folded by syn-anatectic deformation and the fold axial plane (S_2) is generally E–W striking and S-dipping. These observations indicate that the protolith of the migmatite may have undergone near southward movement which occurred simultaneously with the anatexis process. This southward kinematics is consistent with a forearc thrust scenario. We added these geological observations and related figures into the revised manuscript.

Zhao, J.H., Zhou, M.F., Yan, D.P., Zheng, J.P., Li, J.W., 2011. Reappraisal of the ages of Neoproterozoic strata in South China: No connection with the Grenvillian orogeny. *Geology* 39, 299–302.

Cawood, P.A., Zhao, G.C., Yao, J.L., Wang, W., Xu, Y.J., Wang, Y.J., 2018. Reconstructing South China in Phanerozoic and Precambrian supercontinents. *Earth-Science Rev.* 186, 173–194.

Shu, L.S., Yao, J.L., Wang, B., Faure, M., Charvet, J., Chen, Y., 2021. Neoproterozoic plate tectonic process and Phanerozoic geodynamic evolution of the South China Block. *Earth-Science Rev.* 216, 103596.

Shu, L.S., Zhou, G.Q., Shi, Y.S., Yin, J., 1994. Study of the high-pressure metamorphic blueschist and its Late Proterozoic age in the Eastern Jiangnan Orogenic Belt. *Chin. Sci. Bull.*, 39, 1200–1204.

Sun, Z.M., Wang, X.L., Qi, L., Zhang, F.F., Wang, D., Li, J.Y., Yu, M.G., Shu, X.J., 2018. Formation of the Neoproterozoic ophiolites in southern China: New constraints from trace element and PGE geochemistry and Os isotopes. *Precamb. Res.* 309, 88–101

Li, J.Y., Wang, X.L., Wang, D., Du, D.H., Yu, J.H., Gu, Z.D., Huang, Y., Li, L.S., 2021. Pre-Neoproterozoic continental growth of the Yangtze Block: From continental rifting to subduction–accretion. *Precamb. Res.* 355, 106081.

Wang, X.L., Zhou, J.C., Qiu, J.S., Gao, J.F., 2004. Geochemistry of the Meso- to Neoproterozoic basic-acid rocks from Hunan Province, South China: implications for the evolution of the western Jiangnan orogen. *Precambrian Res.* 135, 79–103.

Wang, X.L., Zhou, J.C.*, Qiu, J.S., Jiang, S.Y., Shi, Y.R., 2008. Geochronology and geochemistry of Neoproterozoic mafic rocks from western Hunan, South China: implications for petrogenesis and post-orogenic extension. *Geological Magazine* 145, 215–233.

Wang, X.L., Zhou, J.C., Griffin, W.L., Zhao, G.C., Yu, J.H., Qiu, J.S., Zhang, Y.J., Xing, G.F., 2014. Geochemical zonation across a Neoproterozoic orogenic belt: Isotopic evidence from granitoids and metasedimentary rocks of the Jiangnan orogen, China. *Precambrian Research* 242, 154–171.

Zhao, J.H., Li, Q.W., Liu, H., Wang, W., 2018. Neoproterozoic magmatism in the western and northern margins of the Yangtze Block (South China) controlled by slab subduction and subduction-transform-edge-propagator. *Earth-Science Rev.* 187, 1–18.

Wang, X.L., Zhou, J.C., Griffin, W.L., Zhao, G.C., Yu, J.H., Qiu, J.S., Zhang, Y.J., Xing, G.F., 2014. Geochemical zonation across a Neoproterozoic orogenic belt: Isotopic evidence from granitoids and metasedimentary rocks of the Jiangnan orogen, China. *Precambrian Research* 242, 154–171.

Zheng, Y.-F., Wu, R.-X., Wu, Y.-B., Zhang, S.-B., Yuan, H.L., Wu, F.-Y., 2008. Rift melting of juvenile arc-derived crust: geochemical evidence from Neoproterozoic volcanic and granitic rocks in the Jiangnan Orogen, South China. *Precamb. Res.* 163, 351–383.

Gao, R. Chen, C. Wang, H. et al. *SINOPROBE deep reflection profile reveals a Neo-Proterozoic subduction zone beneath Sichuan Basin. Earth Planet. Sci. Lett.*, 454 (18) (2016), pp. 86-91.

Jia D., Wei G., Chen Z.X., Li B.L., Zen Q., Yang G. *Longmen Shan fold-thrust belt and its relation to the western Sichuan Basin in central China: new insights from hydrocarbon exploration. Am. Assoc. Petrol. Geol. Bull.*, 90 (2006), pp. 1425-1447.

Wang, E. Meng, Q. *Mesozoic and Cenozoic tectonic evolution of the Longmenshan fault belt. Science in China Series D: Earth Sciences*, 52 (2009), pp. 579-592.

Basically, we are shown some zircons whose low delta 18O cores are overprinted by high delta 18O rims formed not long after- 10-20 My later. That is the paper. Is there any other possibility to increase the oxygen via meteoric water fluxing at depth? It's marginally possible: some work done in the 1990s on core complex 18O (Greg Holk, others) show that meteoric water can be sent down to great depths along normal faults where it can damage the oxygen isotopic signature of minerals like quartz (and presumably zircon as well).

We agree that downward infiltration of meteoric water may be an important mechanism to introduce oxygen isotopic heterogeneity to the deep crust, although it would be difficult because of deep crust at normal thermal gradient could not support a deep migration of meteoric water. When it happens, this process would likely cause water-rock interaction at high temperatures and impart low $\delta^{18}\text{O}$ to the rocks, as has been clearly seen in the lower oceanic crust (Roberts and Spencer, 2015). So, this mechanism may not explain the high $\delta^{18}\text{O}$ recorded by the zircon rims of this study, which requires water-rock interactions at low temperatures, most likely under surface conditions. Holk GJ and Taylor HP (1997, 2000) suggested that the ^{18}O depletion of feldspar ($\delta^{18}\text{O} = -4.0\text{‰}$) within the imbricate thrust zone and in the vicinity of the Columbia River fault was the result of late-stage meteoric-hydrothermal exchange associated with detachment faulting. Their work also indicates that meteoric-hydrothermal activity will clearly lower $\delta^{18}\text{O}$ values of the mineral. On the other hand, the maximum penetration depth of meteoric water generally ranges from 5 to 18 km (Menzies et al., 2014 and references therein), which is lower than the depth of anatexis (~ 18 to 24 km) calculated for our migmatites.

Roberts, N.M.W., & Spencer, C.J., 2015. The zircon archive of continental formation through time. Geol. Soci. Lon. Spec. Pub. 389, 197–225.

Holk GJ, Taylor HP, 1997. 18O/16O homogenization of the middle crust during anatexis: The Thor-Odin metamorphic core complex, British Columbia. Geology 25, 31–34

Holk GJ, Taylor HP, 2000. Water as a Petrologic Catalyst Driving 18O/16O Homogenization and Anatexis of the Middle Crust in the Metamorphic Core Complexes of British Columbia. International Geology Review 42, 97–130

Menzies, C.D., Teagle, D.A.H., Craw, D., Cox, S.C., Boyce, A.J., Barrie, C.D., and Roberts, S., 2014. Incursion of meteoric waters into the ductile regime in an active orogen. Earth and Planetary Science Letters, 399, 1–13.

Lots of similar work on the cannibalization topic has been done in areas that have a better known geologic control, such as the western N American Cordillera (the POR work started in the 1970s

-Platt, England, many more; the newer work based on zircons – Jacobson, Grove, Barth, Kidder, Ducea in California, Matzel, Gordon, etc etc in the Pacific NW). Some of it is acknowledged here, some other relevant papers, not. There, hypotheses such as forearc thrusting can be put to test. Incidentally, forearc thrusting is rarely the explanation in north America, and structural evidence for that mechanism is most compelling in the central Andes. That vast literature shows that there are several sources for recycled materials in an arc, forearc “thrusting” being just one of them. Ultrashallow subduction does it too at high rates (centimeters per year, see the Southern Californian example – Pelona etc), thrusting from the backarc as well (at mm/year) and vertical transport within the arc (the downward flow model of Saleeby or Paterson, others) is also an option, not mentioned here. Rate constraints provided here are minimalistic (at least a few mm/yr) so that alone cannot distinguish between different hypotheses. A careful look at inherited, older than the arc zircons (if they exist) could distinguish between fore and back arc origins. Downward flow is hard to estimate without a good geologic context.

We agree that both forearc and backarc thrusting can transport surface materials to the deep arc crust, and both have been suggested for N American Cordillera (e.g., DeCelles et al., 2009; Pearson et al., 2017; Sauer et al., 2017; Ducea and Chapman, 2018). Ultrashallow subduction is a type of thrusting in essence.

The core zircons in the studied migmatites have ages of 930~820 Ma. The absence of considerable Pre-Neoproterozoic zircon core domains or zircon xenocrysts seems to preclude the correlation with backarc strata. In contrast, the 930~820 Ma zircon core domains mainly have positive $\epsilon_{\text{Hf}}(t)$ and mantle-like oxygen isotope compositions, supporting an origin of the nearby Neoproterozoic juvenile arc materials. We thus suggest the precursor of these migmatites were transported downward via forearc thrust channels. In addition, the syn-anatexis kinematics observed in the field (as discussed above) is also consistent with a forearc-side thrusting scenario.

We also agree that downward flow of wall rocks during magma ascent may be an important crustal recycling mechanism in magmatic arcs. In the revised manuscript, we acknowledged the possibility of downward flow in lines 201~204. Most downward transport of host rock in plutonic systems occurs within narrow aureoles and/or in the ascending magma column (Paterson and Farris, 2006). In this study, it's hard to evaluate the role of downward flow in transporting surface materials to the deep crust, but we think it is less likely because (1) host rock xenoliths were not seen in the plutonic rocks, and (2) vertical flow foliation or lineation is absent in the host rocks and migmatites.

DeCelles, P. G., Ducea, M. N., Kapp, P. & Zandt, G. Cyclicity in Cordilleran orogenic systems. Nature Geoscience. 2, 251–257 (2009).

Pearson, D.M. et al. 2017. Sediments underthrusting within a continental magmatic arc: Coast Mountains batholith, British Columbia. Tectonics 36, 2022-2043.

Ducea, M. N. & Chapman, A. D. Sub-magmatic arc underplating by trench and forearc materials in shallow subduction systems: A geologic perspective and implications. Earth-Sci. Rev. 185, 763–779 (2018).

Paterson, S.R. & Farris, D.W., 2006. Downward host rock transport and the formation of rim monoclines during the emplacement of Cordilleran batholiths. Earth Environ. Sci. Trans. Royal Soci. Edinb. 97, 397-413.

Sauer, K. B., Gordon, S. M., Miller, R. B., Vervoort, J. D. & Fisher, C. M. Transfer of metasupracrustal rocks to midcrustal depths in the North Cascades continental magmatic arc, Skagit Gneiss Complex, Washington. Tectonics 36, 3254–3276 (2017).

Some data presented here don't advance the paper's primary argument. For example, Hf isotopes are for the most part useless here. Perplex modeling is only marginally useful because it has been clearly documented these rocks were last seen as migmatites at high temperatures, so yeah, they were part of a hot geotherm. Some references, like the one to Penniston Dorland on what is a hot or cold (subduction) gradient are weird. That is textbook stuff, if anything you can refer to early classic papers by Peacock or Phil England.

Thanks for your suggestion. The reference to Penniston Dorland has been replaced by the work by Peacock (1990). P-T-t paths predicted by Peacock (1990) for subducted slab top are displayed for different convergence rates (3cm/yr and 10cm/yr), positions in crust and amounts of previously subducted lithosphere. And the P-T-t path followed by the subducting oceanic crust is primarily a function of the amount of lithosphere previously subducted and not the convergence rate. Therefore, it can be inferred that the slab tops are cold, with their $dT/dP \approx 34$ °C/kbar, inconsistent with the hot-geotherm anatexis condition in this study.

We would respectively argue that Hf isotopes are useful here in that the Hf data allow us to evaluate whether the sedimentary precursors were derived from the juvenile arc crust itself or the ancient continental crust. This point is also indicated in the last response to your comments.

Peacock, S. M., Numerical simulation of metamorphic pressure-temperature-time paths and fluid production in subducting slabs, Tectonics, 9, 1197 – 1211, 1990.

I don't have the time to go through the details of the writing, will only point out that duration in geology is My or Myr as opposed to absolute age which one writes as Ma. I am fine with the writing, anybody can understand the message.

Corrected. Thanks for pointing this out.

Is this worthy of being published in Nature Comm? Only an editor can judge the appeal of this manuscript to their journal. I liked the paper and would publish it if it were up to me, but would have also liked to see more details from within zircons (where the actual data comes from), such as how big the rims are, do they show metamorphic U/Th, what are Ti temperatures across grains, trace chemistry etc. All the paper's punch rests on investigations within grain (age and oxygen) anyway. There is no geology here and for the record, thrust faults are mapped/inferred at a whole different scale with totally different tools. Thrust faults cannot be deduced from "within grain" studies but certain geochemical constraints on the rock cycle can be unraveled at that scale.

We have now supplemented our original dataset with zircon trace element composition data. The detailed representative cathodoluminescence (CL) images are also attached as a separate file in which the beam size and dating positions are shown in Supplementary Fig. 8, 9 clearly. Since the zircons previously analyzed for U-Pb-Hf-O isotopes have been largely consumed, we carried

out the trace element analyses using other clean rim and core zircons. We put the trace element plots below. The core and overgrowth rim domains of these zircons are not systematically different. We substantially expanded geologic background. And our new field observations also give more evidence for thrusting speculation.

1) zircon size:

- a) 16YX-1-1, whole zircon grains size: mostly 40~90 μm ; Core zircon size: 20~60 μm ;
- b) 16YX-4-1, whole zircon grains size: 90~170 μm ; Core zircon size: 50~120 μm ;
- c) 15PG-32-1, whole zircon grains size: 50~120 μm ; Core zircon size: 30~100 μm .

2) zircon Th/U ratios:

Nearly all rim zircon domains have Th/U ratios < 0.1 (metamorphic or anatectic), and all core zircon domain have Th/U ratios > 0.3 (igneous).

3) zircon trace chemistry :

Core and rim zircon grains generally show similar trace element patterns (chondrite-normalized). They have low concentrations of LREEs, positive Ce and negative Eu anomalies. The HREE display an increasing enrichment (Gd 10 times, Lu 1000 times C1) with decreasing ionic radii. Concentrations of Nb and Ta are low in zircon.

4) Ti in zircon temperature (calculation method from Ferry and Watson, 2007):

From core to rim zircon, Ti-in-zircon temperatures show a slight descend in samples 16YX-1-1 and 16YX-4-1, but a slight increase in sample 15PG-32-1.

16YX-4-1: Core Zircon $T_{\text{Ti-in-zircon temperature}} = 710 \pm 14$ (weighted mean, 2s, n=18, MSWD=2.73);

Rim Zircon $T_{\text{Ti-in-zircon temperature}} = 688 \pm 22$ (weighted mean, 2s, n=11, MSWD=2.21)

16YX-1-1: Core Zircon $T_{\text{Ti-in-zircon temperature}} = 775 \pm 42$ (average, 2s, n=2);

Rim Zircon $T_{\text{Ti-in-zircon temperature}} = 764 \pm 16$ (weighted mean, 2s, n=8, MSWD=1.07)

15PG-32-1: Core Zircon $T_{\text{Ti-in-zircon temperature}} = 706 \pm 80$ (average, 2s, n=5);

Rim Zircon $T_{\text{Ti-in-zircon temperature}} = 768 \pm 30$ (average, 2s, n=4)

Ferry, J.M., Watson, E.B., 2007. New thermodynamic models and revised calibrations for the Ti-in-zircon and Zr-in-rutile thermometers. *Contributions to Mineralogy and Petrology* 154, 429–437.

Reviewer #3 (Remarks to the Author):

Dear Editor,

Below is my review of *Rapid endogenic rock recycling in magmatic arcs* by Wang et al. The paper is very nicely written and provides important new data supporting rapid incorporation of supracrustal materials into magmatic systems. I think the greatest weakness of the paper is the reliance of thrust fault networks on transporting sedimentary material into the “magmatic window”. It would be useful to provide some geologic evidence for this phenomena happening in active continental arcs.

Thanks for your valuable comments. The lack of sufficient geologic context was also pointed out by the first reviewer. As responded to the comments of the first reviewer, we made another field trip to the study area to search for additional geologic evidence for the Neoproterozoic thrust.

And based on our new observations, we substantially fleshed out the “Geologic setting and samples” section. In short:

- 1) the migmatites were intruded by dioritic plutons (~820 Ma, arc-like geochemistry) and numerous mafic dikes (~780 Ma and ~270 Ma, weak deformation);
- 2) the stromatic migmatites with melts generated along the foliation planes were extensively outcropped in the field;
- 3) syn-anatectic deformation structure within the migmatite indicates a possible southward movement, consistent with forearc thrust scenario;
- 4) the major Cenozoic extrusion and thrust process uplifted the ancient rocks to the surface;
- 5) the Neoproterozoic tectonic imprints were largely overprinted by younger tectonic events.

Below are a few additional requested revisions that would improve the paper.

1) Zircon methodologies are insufficient and need to be expanded. The ICP-MS data are mostly reported in a palatable fashion however the analytical methodology is not adequately explained in the supplementary material.

Thanks for your valuable advice. We have now significantly expanded the zircon methodologies in the “Methods” section.

a. Although the laboratories have been generating high quality data for many years for the sake of thoroughness and reproducibility, the authors need to provide the ‘metadata’ associated with their analytical sessions (for example laser energy, gas flows, etc.). There are many specific analytical conditions of ICP-MS instruments that change within each analytical session that are important to report. Specific to the ICP-MS data the authors should follow the recommendations outlined by Horstwood et al. (2016 GGR) with respect to reporting of analytical metadata.

Thanks for this suggestion. The ICP-MS instruments were used to analyze zircon Hf isotopic compositions in this study. In the revised “Methods” section, we followed the suggestions by the reviewer and revised the analytical section following the style of Horstwood et al. (2016 GGR) in the following aspects: (1) laboratory name; (2) sample type and preparation; (3) CL imaging (instrument and conditions); (4) Make, Model and type; (5) ablation duration, size, pulse rate and energy density; (6) carrier gas; (7) sampling mode; (8) data processing and reference material information and (9) data quality.

b. The authors should also provide information regarding the stable isotope ratios for the zircon Hf data ($^{180}\text{Hf}/^{177}\text{Hf}$ and $^{179}\text{Hf}/^{177}\text{Hf}$) to demonstrate the instrument was working properly. Without these it is impossible to determine whether the analyses are valid. This is common practice and these data should be provided. To make the authors lives easier, they can simply follow the workflow provided by Spencer et al. (GSF 2019).

We agree and provided information on the stable Hf isotope ratios (i.e., $^{178}\text{Hf}/^{177}\text{Hf}$ and $^{180}\text{Hf}/^{177}\text{Hf}$) have been demonstrated in supplementary Table 3 following the workflow provided

by Spencer et al. (2019). Analyzed $^{176}\text{Hf}/^{177}\text{Hf}$, $^{178}\text{Hf}/^{177}\text{Hf}$ and $^{180}\text{Hf}/^{177}\text{Hf}$ ratios for two reference zircons (Mudtank and 91500) are illustrated in text and below:

c. The authors should report the value, uncertainty, and MSWD of the secondary standard material for UPb and Hf analyzed to confirm correct normalization. The authors say they used three different standards but did not indicate which standard is the primary (self-normalized) and secondary (used to assure normalization is accurate).

We have now provided the value, uncertainty, and MSWD of the secondary standard zircon analyses in the revised manuscript. For zircon U-Pb analysis, zircon Plešovice was the primary standard and zircon Qinghu served as the secondary standard. For zircon oxygen analysis, zircon Penglai was the primary standard while zircon Qinghu was the secondary standard. For zircon Hf isotopes, zircon Mudtank and 91500 served as the secondary standard material. The value, uncertainty and MSWD of the secondary materials for U-Pb-O-Hf analyses are listed below.

1) zircon U-Pb isotope: Qinghu, $^{206}\text{Pb}/^{238}\text{U}$ age = 159 ± 2 Ma (n=11; MSWD = 0.72, single population);

2) zircon O isotope: Qinghu, $\delta^{18}\text{O} = 5.39 \pm 0.08$ ‰ (n=17; MSWD = 1.3, single population);

3) zircon Hf isotope:

a) Mudtank: $^{176}\text{Hf}/^{177}\text{Hf} = 0.282487 \pm 0.000008$
(n=14; MSWD =3.6, over dispersion)

$^{178}\text{Hf}/^{177}\text{Hf} = 1.467216 \pm 0.000023$
(n=13; MSWD =4, over dispersion)

$^{180}\text{Hf}/^{177}\text{Hf} = 1.886871 \pm 0.000038$
(n=13; MSWD =2.1, over dispersion)

b) 91500: $^{176}\text{Hf}/^{177}\text{Hf} = 0.282295 \pm 0.000008$
(n=13; MSWD =2.2, over dispersion)

$^{178}\text{Hf}/^{177}\text{Hf} = 1.467207 \pm 0.000017$
(n=13; MSWD = 1.6, single population)

$$^{180}\text{Hf}/^{177}\text{Hf} = 1.886857 \pm 0.000042$$

(n=13; MSWD =2, over dispersion)

2) *It is not clear how the authors define concordance. Many workers use an arbitrary discordance filter of 10% discordance, but this is unscientific and if the authors used an arbitrary discordance filter they need to justify this with a sensitivity analysis or alternatively they can simply use the uncertainty of the analysis to define whether an analysis is statistically “concordant” or “discordant”. Spencer et al. (2016 Geoscience Frontiers) evaluate the issue of discordance in conjunction with uncertainty. This is simply done by evaluating whether a given analysis is within uncertainty of the concordia. If an analysis has a centroid that is ‘discordant’ but overlaps the concordia within uncertainty, then there is a probability the analysis is actually concordant despite centroid discordance. I urge the authors to re-evaluate the issue of discordance and would recommend against publication of an arbitrary discordance distinction.*

We followed the method of Spencer et al. (2016 Geoscience Frontiers) and re-evaluated all analyses using the criteria below. First, only the analyses with overlapping U-Pb ages (within 2σ) were used. Second, we use $^{238}\text{U}/^{206}\text{Pb}$ age as the “best age” due to its greater precision. Then, we further filtered out the analyses with >10% discordance which is appropriate for Neoproterozoic zircon.

3) *As a minor point, zircon is a mineral species names and should not be used in the plural sense. The singular is used when writing about a mineral (e.g. titanite) is that it is formally considered a species. Species names are not pluralized in English. For example, *Vulpes vulpes* = red fox. And while we might refer to “foxes in the forest”, we wouldn't refer to “*Vulpes vulpes-es* in the forest”. Confusion can arise when a common mineral name is also a mineral group name. In contrast “garnets” can refer to different garnet species [spessartine, pyrope, etc.].*

Corrected. Thanks for the suggestion.

REVIEWERS' COMMENTS

Reviewer #1 (Remarks to the Author):

I am happy with the detailed responses and the thorough attention to detail that the manuscript now shows. I think it should be published as is. I look forward to see this published. Sincerely Mihai Ducea

Reviewer #3 (Remarks to the Author):

Dear Editor,

It was my pleasure to read over this revised manuscript along with the authors' responses to the critiques and queries. While I am satisfied with the revisions that have been made, I was slightly dismayed that the authors claim to have revised the manuscript based upon previous work, but few if any of the references discussed in their response to the reviewers were included in the final manuscript. If the authors feel the manuscript was improved by the citations provided by the reviewers, it seems reasonable that those references would be cited in the actual manuscript aside from simply paying lip service to them in the responses to the reviewers.

REVIEWERS' COMMENTS

Reviewer #1 (Remarks to the Author):

*I am happy with the detailed responses and the thorough attention to detail that the manuscript now shows. I think it should be published as is. I look forward to see this published. Sincerely
Mihai Ducea*

Thank you for your in-deep review.

Reviewer #3 (Remarks to the Author):

Dear Editor,

It was my pleasure to read over this revised manuscript along with the authors' responses to the critiques and queries. While I am satisfied with the revisions that have been made, I was slightly dismayed that the authors claim to have revised the manuscript based upon previous work, but few

if any of the references discussed in their response to the reviewers were included in the final manuscript. If the authors feel the manuscript was improved by the citations provided by the reviewers, it seems reasonable that those references would be cited in the actual manuscript aside from simply paying lip service to them in the responses to the reviewers.

We apologize for missing these references in our last revision, which we have now added to the manuscript.

Specifically:

- 1) lines 61-63, we cited previous work by Zhao et al. (2011), Cawood et al. (2018) and Shu et al. (2021) to claim that the Yangtze Block placed a marginal position in Rodinia supercontinent and underwent long-term evolution in an active-continent-marginal setting during Neoproterozoic;
- 2) in the legend of Figure 1 (line 124), we cited the work from Spencer et al. (2016) to clarify the concordant criteria on zircon U-Pb age.
- 3) lines 190-198, we cited the work from Roberts and Spencer. (2015) and Menzies et al. (2014) to demonstrate that downward infiltration of meteoric water may not explain the high $\delta^{18}\text{O}$ recorded by the zircon rims of this study.